evolution/ecology

telomere dynamics, sexual size dimorphism, heterogametic sex, ageing, longevity, life-history strategies

**Author for correspondence:**
Florentin Remot
e-mail: florentin.remot@gmail.com

# No sex differences in adult telomere length across vertebrates: a meta-analysis

Florentin Remot[1], Victor Ronget[1], Hannah Froy[2,3], Benjamin Rey[1], Jean-Michel Gaillard[1], Daniel H. Nussey[2] and Jean-François Lemaître[1]

[1]Université de Lyon, Université Lyon 1, CNRS, Laboratoire de Biométrie et Biologie Evolutive, UMR5558, F-69622 Villeurbanne, France
[2]Institute of Evolutionary Biology, University of Edinburgh, Edinburgh EH9 3FL, UK
[3]Centre for Biodiversity Dynamics, Norwegian University of Science and Technology, Trondheim, Norway

FR, 0000-0001-8999-925X; BR, 0000-0002-0464-5573;
DHN, 0000-0002-9985-0317; J-FL, 0000-0001-9898-2353

In many mammalian species, females live on average longer than males. In humans, women have consistently longer telomeres than men, and this has led to speculation that sex differences in telomere length (TL) could play a role in sex differences in longevity. To address the generality and drivers of patterns of sex differences in TL across vertebrates, we performed meta-analyses across 51 species. We tested two main evolutionary hypotheses proposed to explain sex differences in TL, namely the heterogametic sex disadvantage and the sexual selection hypotheses. We found no support for consistent sex differences in TL between males and females among mammal, bird, fish and reptile species. This absence of sex differences in TL across different classes of vertebrates does not support the heterogametic sex disadvantage hypothesis. Likewise, the absence of any negative effect of sexual size dimorphism on male TL suggests that sexual selection is not likely to mediate the magnitude of sex differences in TL across vertebrates. Finally, the comparative analyses we conducted did not detect any association between sex differences in TL and sex differences in longevity, which does not support the idea that sex differences in TL could explain the observed sex differences in longevity.

## 1. Introduction

Telomeres are DNA structures composed of non-coding sequences that are repeated in tandem at the extremity of linear chromosomes. Telomeres are essential to the maintenance of genomic integrity and

protect coding DNA against two major deleterious processes: 'the end replication problem' and oxidative damage [1]. In the absence of telomerase, an enzyme that acts to elongate telomeres, telomeres unavoidably shorten with each DNA replication cycle. Critically short telomeres are associated with genomic instability (e.g. chromosome fusions, chromosome breaks), cell cycle arrest and can trigger apoptosis. The shortening of telomere sequences with increasing age is now considered as a key physiological mechanism underlying ageing [2]. While a causal association among telomere length (TL), age-specific diseases and survival prospects remains uncertain [3], telomere dynamics have been associated with several age-related disorders such as cardiovascular diseases [4] in humans. In addition, shorter telomeres are associated with a higher mortality risk in humans [5] and in several populations of vertebrates [6].

Women and men show striking differences in adult health and lifespan, the shorter lifespan of men being a particularly robust feature across populations worldwide [7]. Among the many complex genetic and physiological factors that might govern those sex differences, a possible role played by sex-specific telomere dynamics has been proposed [8]. Recently, a meta-analysis on sex differences in TL in humans (compiling 40 studies) revealed that adult women have on average slightly longer telomeres than adult men (i.e. 176 bp longer in women than in men of similar age). However, this study also revealed high heterogeneity in the magnitude and direction of the sex differences observed across studies ($I^2 = 91.4\%$) [9]. So far, most human studies have been cross-sectional and have focused on middle-aged or elderly individuals. Thus, it has not been possible to determine whether average sex differences in adult TL result from differences determined genetically or in early life and maintained through adulthood, or from differences in the rate of telomere attrition (i.e. telomeres shortening faster in men over time), or both. A recent study based on umbilical cord blood found 144 bp longer telomeres in girls than in boys [10], suggesting that sex differences in human TL are already present at birth. However, these results are inconsistent with previous observations on humans that show no gender difference in TL at the start of life (at birth and *in utero*) [8].

Until now, two main evolutionary hypotheses have been proposed to explain why TL could differ between sexes [8]. First, the heterogametic disadvantage hypothesis predicts that the heterogametic sex (e.g. XY males in mammals) should have shorter telomeres than the homogametic sex (e.g. females XX in mammals) if the Y chromosome contains fewer telomere maintenance alleles [8]. Moreover, if males carry a deleterious mutation in telomere maintenance alleles on the X chromosome, this mutation will always be expressed in males while in females it can be compensated by the second X chromosome [11]. Therefore, the heterogametic disadvantage hypothesis is not based on a difference in telomere length between the X and Y chromosomes, but on a difference in gene content, especially those involved in telomere maintenance. For instance, the DKC1 gene that codes for dyskerin, a protein involved in the stability of telomerase, is located on the X chromosome [1]. Mutations of this gene induce dyskeratosis congenita whose symptoms resemble premature ageing and critically short telomeres. In birds, due to their ZW sex-determination system (i.e. males are ZZ and females are ZW), this hypothesis predicts shorter telomeres in females, assuming that telomere maintenance alleles are located on the homogametic sex chromosome (Z chromosome). In reptiles and fish, predictions are more complicated and will depend on the sex-determination system of each species. This hypothesis is indirectly supported by a recent meta-analysis, which showed that in vertebrates the heterogametic sex lives shorter than the homogametic sex [12]. Second, the sexual selection hypothesis posits that the sex growing faster and maintaining a larger body mass throughout the lifetime should display shorter telomeres due to increased cell replication during the growth phase and the regeneration of tissues along the life course [13,14]. Mammals, for instance, tend to show often a strong male-bias in body size (associated with polygynous mating systems) [15], meaning that, following the sexual selection hypothesis, males should have shorter telomeres than females. However, unlike the heterogametic disadvantage hypothesis, the sexual selection hypothesis makes predictions within groups as well as across groups because variation in sexual size dimorphism occurs both among and within vertebrate groups. So far, the vast majority of research on the genetic and physiological determinants of sex differences in ageing has focused on laboratory organisms [7] (but see [11]). To fully understand the universal mechanisms that modulate ageing, we need to know whether mechanisms that have been suggested to contribute to ageing, such as telomere dynamics, are shared between different lineages or if these mechanisms are species-specific [16]. To evaluate the role of evolutionary forces in shaping sex differences in vertebrate TL, it is necessary to test these hypotheses across a wide range of species displaying contrasted life-history strategies.

A previous qualitative review of sex differences in TL across animals suggested that telomeres may be longer in females than in males in mammals, but not in birds [8]. However, these observations were made on a limited number of species (i.e. 20 species of birds and 5 species of mammals), with a bias toward laboratory rodents in mammals. Laboratory rodents are known to display unusual telomere dynamics, which are

neither representative of mammals in general nor their wild counterpart (i.e. telomeres in laboratory mice are known to be particularly long compared with those in wild-derived mice, see [17,18]). They may therefore provide only limited insight into the evolutionary forces that have shaped sex differences in TL across vertebrates. The recent surge of studies investigating telomere dynamics in wild populations [6,19,20] now provides the necessary material to test whether consistent sex differences in mean TL actually occur across vertebrates and to quantify the magnitude of these differences. In this article, we used a meta-analytic approach to examine sex differences in mean TL across non-human vertebrates and to test the main evolutionary hypotheses proposed to explain such sex differences. According to the heterogametic sex hypothesis, we predicted that the heterogametic sex should have shorter telomeres than the homogametic sex. According to the sexual selection hypothesis, we expected that the magnitude of sex differences in mean TL would be positively associated with the intensity of sexual size dimorphism. Finally, we examined whether sex differences in mean TL could explain differences in life expectancy between the sexes, which are widely observed among vertebrates [11,21]. We predicted that, across species, the longer lived sex should have longer telomeres.

# 2. Material and methods

## 2.1. Literature survey

Relevant articles were collected by entering the following keywords 'telom* NOT (clinic* OR hospital)' in the 'topic' window of the ISI Web of Science database. The keywords telom* allowed us to find articles not only about telomeres but also about telomerase, and the use of 'NOT (clinic* OR hospital)' enabled us to exclude many studies exclusively focused on humans. The literature search was restricted to the following Web of Science Categories: 'Evolutionary biology', 'Marine freshwater biology', 'Multidisciplinary sciences', 'Geriatrics gerontology', 'Physiology', 'Zoology', 'Environmental sciences', 'Fisheries', 'Ecology', 'Agriculture dairy animal sciences' and 'Veterinary sciences'. The search was conducted in January 2019. Using this protocol, 5029 articles were gathered. We performed a first selection based on the title and the abstract to exclude single sex or human studies. All the remaining articles ($N = 324$) were then read in full to exclude studies according to a predefined set of criteria (figure 1). In experimental studies, we excluded the 'treatment group' because the treatment can influence telomere dynamics [19], potentially in a sex-specific way, but we retained the 'control group'. Similarly, we excluded studies that measured telomeres in cell cultures, in laboratory strains or in domesticated species because these measurements were unlikely to reflect telomere length in natural conditions [17]. We also excluded data on neonates (i.e. before fledging in birds, before weaning in mammals) because the number of newborn individuals (which is higher than older individuals) and the different dynamics of telomere between them [23] may mask sex differences in TL if these differences occur later in life. Finally, we checked the full reference lists of the 324 selected articles and of the two meta-analyses performed on TL across several populations of vertebrates [5,17] to check that we had not missed any relevant article with our literature search protocol. This procedure allowed us to include data from two PhD theses [24,25] in our analyses. It appeared that some datasets were published, partially or entirely, in more than one article (i.e. 12 datasets or subset of datasets were published more than once). In those cases, we selected the article with the largest dataset or if the datasets were identical, the most recent. By applying this research protocol and the exclusion criteria, we obtained a total of 58 articles covering 52 vertebrate species (figure 1).

## 2.2. Extraction of effect sizes

TL was frequently reported as mean absolute or relative TL (depending on the measurement method, see below) with the standard deviation (or the standard error) and the sample size. When the full dataset was available (e.g. data published in research data repositories such as Dryad), the mean and the standard deviation of TL for both sexes were calculated from the raw data. When the TLs were only graphically displayed in the papers, we extracted the data using the WebPlotDigitizer software [26], and then calculated the mean and the standard deviation of TL for each sex separately. When no data were reported in the article and no dataset was publicly available, we requested the following information from the corresponding authors: mean TL, standard deviation and sample size for both sexes.

To be comparable across studies, effect sizes of the mean differences in TL need to be standardized. The most commonly used standardized mean difference is the 'Hedge's g' (or 'Hedge's d') (see [27] for further details), which is not affected by unequal sampling variance. Females were set as the reference

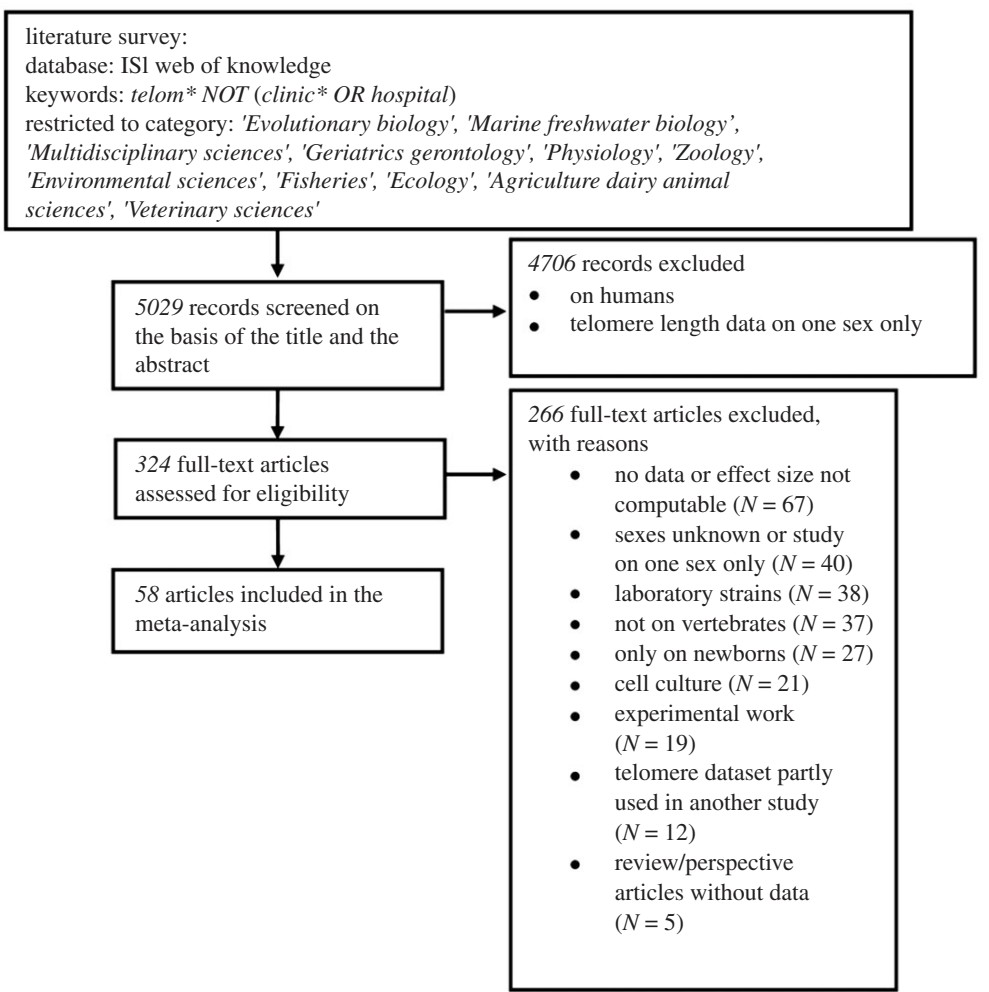

**Figure 1.** PRISMA flow diagram (from the PRISMA statement, [22]) for the meta-analysis.

group. Therefore, a positive value of $g$ means that males have longer telomeres than females, whereas a negative value of $g$ means that females have longer telomeres than males. For the meta-analysis, the interpretation of the effect size followed the Cohen's rule of thumb [28] stating that $g$ absolute values of 0.2, 0.5 and 0.8 represent low, medium, and large differences, respectively (effects size of each study are summarized in table 1).

## 2.3. Additional information extracted from the articles or the literature

All information required for the identification of the paper (title, first author, year of publication, journal, species and population studied) was recorded. We also extracted information regarding the method of TL measurement (e.g. TRFS for telomere restriction fragment followed by a Southern blot, TRFI for telomere restriction fragment followed by an in-gel hybridization and quantitative PCR), the biological tissue sampled to measure telomeres (e.g. red blood cell, white blood cell or other tissues), and whether telomeres were measured in free-living or captive animals. We also created a data extraction quality index (whether the effect size was computed from a graph [i.e. low-quality index] or directly from data in the article [i.e. high-quality index]). To assess the factors of variation in the magnitude of sex differences in TL, we then searched for the following information in the literature for the set of the 52 vertebrate species included in the meta-analysis: female age at first reproduction (log-transformed), mean adult body mass (log-transformed), mating system and mean annual adult survival for both sexes. All variables are fully described in electronic supplementary material, Methods and Results and were included as moderators in the meta-regression separately (i.e. subset analysis) to avoid over-fitting the model.

**Table 1.** Summary of the statistics and of the potential driving factors compiled in the meta-analysis. For each study, the sex difference in TL ($g$), its variance and its sample size are reported. The tissue sampled is reported as RBC for red blood cells, WBC for white blood cells and other for all other types of tissue. The method of TL measurement is reported as TRFS for telomere restriction fragment followed by a Southern blot, TRFI for telomere restriction fragment followed by an in-gel hybridization, and qPCR. See electronic supplementary material, Methods and Results for further information.

| species | study | $g$ | variance | sample size | tissue | method |
|---|---|---|---|---|---|---|
| **BIRDS** | | | | | | |
| *Acrocephalus arundinaceus* | [29] | −0.086 | 0.059 | 68 | RBC | qPCR |
| *Acrocephalus sechellensis* | [30] | 0.174 | 0.005 | 837 | RBC | qPCR |
| *Aptenodytes patagonicus* | [31] | 0.115 | 0.08 | 50 | RBC | qPCR |
| *Aptenodytes patagonicus* | [32] | −0.158 | 0.038 | 106 | RBC | qPCR |
| *Aptenodytes patagonicus* | [33] | −0.094 | 0.027 | 146 | RBC | qPCR |
| *Branta leucopsis* | [34] | 0.678 | 0.141 | 34 | RBC | TRFS |
| *Calidris alpina* | [35] | −0.177 | 0.178 | 24 | RBC | TRFI |
| *Corvus monedula* | [36] | 0.242 | 0.084 | 48 | RBC | TRFI |
| *Diomedea exulans* | [37] | 0.423 | 0.068 | 60 | RBC | TRFS |
| *Fregata minor* | [38] | −0.076 | 0.229 | 22 | RBC | TRFI |
| *Hirundo rustica* | [39] | 0.17 | 0.031 | 130 | RBC | qPCR |
| *Larus crassirostris* | [40] | −0.278 | 0.051 | 82 | RBC | TRFS |
| *Larus crassirostris* | [41] | −0.214 | 0.072 | 72 | RBC | TRFS |
| *Leucophaeus pipixcan* | [42] | −0.437 | 0.205 | 20 | RBC | qPCR |
| *Luscinia svecica* | [43] | −0.15 | 0.042 | 97 | RBC | qPCR |
| *Macronectes giganteus* | [44] | −0.653 | 0.092 | 47 | RBC | TRFS |
| *Macronectes halli* | [24] | −1.016 | 0.124 | 37 | RBC | TRFS |
| *Parus caeruleus* | [45] | 0.384 | 0.079 | 56 | RBC | qPCR |
| *Parus major* | [46] | −0.483 | 0.099 | 42 | RBC | TRFI |
| *Parus major* | [47] | 0.483 | 0.053 | 79 | RBC | qPCR |
| *Parus major* | [47] | 0.018 | 0.054 | 76 | RBC | qPCR |
| *Riparia riparia* | [35] | −0.158 | 0.267 | 16 | RBC | TRFI |
| *Rissa tridactyla* | [48] | 0.457 | 0.111 | 38 | RBC | TRFS |
| *Sterna hirundo* | [49] | 0.139 | 0.017 | 233 | RBC | TRFI |
| *Strigops habroptila* | [50] | 0.6 | 0.062 | 67 | RBC | TRFS |
| *Strix aluco* | [51] | 0.06 | 0.026 | 158 | RBC | qPCR |
| *Tachycineta bicolor* | [52] | 0.187 | 0.057 | 82 | RBC | qPCR |
| *Tachymarptis melba* | [53] | 0.044 | 0.041 | 96 | RBC | qPCR |
| *Taeniopygia guttata* | [54] | −0.307 | 0.229 | 19 | RBC | TRFI |
| *Taeniopygia guttata* | [55] | 0.375 | 0.042 | 99 | RBC | qPCR |
| *Taeniopygia guttata* | [55] | 0.128 | 0.052 | 79 | RBC | qPCR |
| *Thalassarche melanophrys* | [56] | 0.091 | 0.079 | 51 | other | TRFS |
| *Turdus merula* | [57] | −0.206 | 0.102 | 40 | RBC | TRFI |
| *Turdus merula* | [57] | −0.607 | 0.192 | 22 | RBC | TRFI |
| *Uria lomvia* | [58] | 0.33 | 0.235 | 35 | RBC | qPCR |
| *Uria lomvia* | [58] | 0.956 | 0.147 | 47 | RBC | TRFS |
| *Uria lomvia* | [59] | 0.578 | 0.101 | 60 | RBC | TRFS |
| *Vultur gryphus* | [60] | 0.566 | 0.21 | 20 | RBC | qPCR |

(Continued.)

| species | study | *g* | variance | sample size | tissue | method |
|---|---|---|---|---|---|---|
| **MAMMALS** | | | | | | |
| *Capreolus capreolus* | [61] | −0.023 | 0.055 | 73 | WBC | qPCR |
| *Capreolus capreolus* | [61] | −0.154 | 0.061 | 66 | WBC | qPCR |
| *Crocuta crocuta* | [62] | −0.771 | 0.091 | 66 | WBC | TRFI |
| *Elephas maximus* | [63] | −0.335 | 0.077 | 120 | WBC | qPCR |
| *Eliomys quercinus* | [64] | −0.382 | 0.259 | 16 | other | qPCR |
| *Glis glis* | [65] | 0.432 | 0.213 | 20 | other | qPCR |
| *Macaca fascicularis* | [66] | 0.11 | 0.202 | 20 | other | TRFS |
| *Macaca fascicularis* | [66] | −0.186 | 0.192 | 21 | other | TRFS |
| *Macaca fascicularis* | [66] | 0.238 | 0.201 | 20 | other | TRFS |
| *Macaca fascicularis* | [66] | −0.819 | 0.219 | 20 | other | TRFS |
| *Macaca fascicularis* | [66] | −0.615 | 0.221 | 19 | other | TRFS |
| *Macaca fascicularis* | [66] | −0.069 | 0.191 | 21 | other | TRFS |
| *Macaca fascicularis* | [66] | 0.63 | 0.233 | 18 | other | TRFS |
| *Macaca fascicularis* | [66] | −0.088 | 0.191 | 21 | other | TRFS |
| *Macaca fascicularis* | [66] | 0 | 0.202 | 20 | other | TRFS |
| *Macaca fascicularis* | [66] | 0.101 | 0.2 | 20 | other | TRFS |
| *Macaca fascicularis* | [66] | 0.571 | 0.21 | 20 | other | TRFS |
| *Macaca fascicularis* | [66] | 0.345 | 0.314 | 13 | other | TRFS |
| *Macaca mulatta* | [67] | −0.251 | 0.084 | 48 | other | TRFS |
| *Macaca mulatta* | [67] | −0.027 | 0.083 | 48 | WBC | TRFS |
| *Macaca mulatta* | [67] | −0.282 | 0.084 | 48 | other | TRFS |
| *Macaca mulatta* | [67] | −0.14 | 0.084 | 48 | other | TRFS |
| *Mandrillus sphinx* | [68] | 1.284 | 0.04 | 120 | WBC | qPCR |
| *Meles meles* | [69] | 0.13 | 0.011 | 360 | WBC | qPCR |
| *Mus musculus* | [70] | 0.055 | 0.243 | 17 | WBC | qPCR |
| *Mus spretus* | [71] | −0.48 | 0.049 | 84 | other | TRFS |
| *Mus spretus* | [71] | −0.73 | 0.045 | 94 | other | TRFS |
| *Mus spretus* | [71] | −0.681 | 0.039 | 110 | other | TRFS |
| *Mus spretus* | [71] | −0.818 | 0.036 | 127 | other | TRFS |
| *Neophoca cinerea* | [72] | −0.052 | 0.167 | 25 | other | qPCR |
| *Ovis aries* | [73] | −0.024 | 0.01 | 481 | WBC | qPCR |
| **FISH** | | | | | | |
| *Cyprinus carpio* | [74] | 0.133 | 0.056 | 73 | other | qPCR |
| *Menidia menidia* | [75] | −0.074 | 0.173 | 29 | other | qPCR |
| *Menidia menidia* | [75] | −0.082 | 0.136 | 38 | other | qPCR |
| *Pungitius pungitius* | [76] | −0.336 | 0.049 | 83 | other | qPCR |
| *Salmo salar* | [77] | −0.251 | 0.062 | 67 | other | qPCR |
| *Salmo salar* | [78] | 0.437 | 0.047 | 89 | other | qPCR |
| *Heterodontus portusjacksoni* | [25] | −0.495 | 0.214 | 20 | other | aqPCR |
| *Heterodontus portusjacksoni* | [25] | 0.45 | 0.182 | 24 | other | qPCR |
| *Heterodontus portusjacksoni* | [25] | −0.31 | 0.202 | 20 | other | TRFS |

(*Continued.*)

| species | study | g | variance | sample size | tissue | method |
|---|---|---|---|---|---|---|
| **REPTILES** | | | | | | |
| *Alligator mississippiensis* | [79] | 0.131 | 0.167 | 24 | RBC | TRFI |
| *Chlamydosaurus kingii* | [80] | −0.239 | 0.102 | 112 | RBC | qPCR |
| *Lacerta agilis* | [81] | −0.323 | 0.025 | 161 | RBC | TRFS |
| *Liasis fuscus* | [82] | −0.909 | 0.123 | 45 | RBC | TRFS |
| *Thamnophis sirtalis* | [22] | −0.638 | 0.06 | 72 | RBC | qPCR |
| *Zootoca vivipara* | [83] | −0.621 | 0.436 | 10 | RBC | TRFS |
| *Zootoca vivipara* | [83] | 0.254 | 0.403 | 10 | RBC | TRFS |
| *Zootoca vivipara* | [83] | 0.687 | 0.424 | 10 | RBC | TRFS |
| *Zootoca vivipara* | [83] | 0.257 | 0.403 | 10 | RBC | TRFS |
| *Zootoca vivipara* | [83] | 0.591 | 0.417 | 10 | RBC | TRFS |
| *Zootoca vivipara* | [83] | 0.601 | 0.418 | 10 | RBC | TRFS |
| *Zootoca vivipara* | [83] | 0.157 | 0.401 | 10 | RBC | TRFS |
| *Zootoca vivipara* | [83] | −0.216 | 0.402 | 10 | RBC | TRFS |
| *Zootoca vivipara* | [83] | −0.39 | 0.408 | 10 | RBC | TRFS |
| *Zootoca vivipara* | [83] | 0.033 | 0.4 | 10 | RBC | TRFS |

## 2.4. General model

The meta-analytic model was run using R (v. 3.3.1, [84]). To account for the non-independence of observations due to pseudo-replication (multiple effect sizes calculated within the same study, species, population or by the same authors), linear mixed-effects models were run with the package MCMCglmm [85]. As related species are more prone to share common trait values, our meta-analytic models controlled for phylogeny [86]. The website http://www.timetree.org/ [87] provided us with a phylogenetic tree (see electronic supplementary material, figure S1) which we further used to control for phylogenetic relatedness among species. This website gathered data from more than 2000 articles published between 1987 and 2013 to estimate divergence times between species and to construct a phylogenetic tree at the vertebrate level. A covariance matrix among the species was then extracted from the tree.

In the meta-analytic model, effect sizes were entered as the dependent variable and the sampling variance associated with each effect size was also included in the model using the *mev* argument. We included the following variables as random factors: the *phylogeny* (using the variance-covariance matrix), the *population* (to control for the fact that several populations were sometimes sampled within a given species), the *sample* (to control for the fact that effect sizes were measured from the same individuals but with different methods or in different tissues) and the *species* (i.e. to take into account that several effect sizes were measured for a given species and can share biological characteristics that are independent of the phylogenetic relatedness such as the habitat). Since we had no *a priori* information on the parameters of the meta-analytic model, we used a non-informative prior (inverse Wishart prior with $v = 0.02$ and $V = 1$). To test if the prior had any impact on the results, we re-ran the meta-analysis using a new parameter expanded prior for the random effect ($v = 1$, $V = 1$, alpha.mu = 0 and alpha.$V = 1000$). Since our results remained unchanged they did not appear to be sensitive to prior specification. The model was run with 1 000 000 iterations (burn-in = 1000 and thinning interval = 50) twice, and convergence of the model was assessed with the Gelman–Rubin diagnostic [88] using the gelman.diag function in R (cut-off value = 1.1, [89]).

From the meta-analytic model, we computed the posterior distribution of the meta-analysis mean and its highest posterior density credible interval (HPDI) at 95%. The effect size was considered significant if the credible interval did not overlap 0. To quantify the heterogeneity in the data accounting for random factors, the $I^2$ and their HPDI at 95% [86] were calculated for each random factor with $I^2 = 25$, 50 and 75% considered as low, moderate and high heterogeneity, respectively. In

addition, we also computed the $H^2$, which is the percentage of the between-study variance explained by the phylogenetic effect. The proportion of the total variance explained by the between-study variance, $I^2_{tot}$, was also calculated [86]. Finally, to assess whether the effect sizes estimated from populations with small sample size could bias our results, we also ran the meta-analytic model including only estimates that were based on 40 or more data points. Overall, results were qualitatively unchanged (see electronic supplementary material, table S1).

## 2.5. Age-correction of telomere length

Two other meta-analyses (based on the same procedure as described above) were then performed to establish whether age effects biased the results in any way. First, we restricted our meta-analysis to measurements made on adults only to remove the effect of a possible sex difference in telomere shortening of juveniles (i.e. before the age at first reproduction) [23]. This meta-analysis included 46 articles and is referred to as 'meta-analysis 2' below. Second, we performed a separate analysis using age-corrected adult TL (referred to as 'meta-analysis 3' below) to correct for possible sex differences in telomere dynamics over the entire life course [23] and for the fact that TL could have been measured at different ages in males and females. For each dataset in which individual ages were available, we first established the best fitting model describing TL as a function of age by comparing three models: no effect of age, a linear relationship with age and a quadratic relationship with age (see electronic supplementary material, table S2 for the full list of models). We compared models using the Akaike information criterion (AIC [90]), selecting the model with the lowest value. When models were within two AIC units, the simpler model was retained. Then, for each sex, we computed the age-corrected TL as the residuals from the selected model. Like in the general meta-analysis, we computed the effect size as the mean difference of the age-corrected TL of both sexes divided by their pooled standard deviation. This meta-analysis included 30 effect sizes (encompassing 23 articles) and is referred to as 'meta-analysis 3' below.

## 2.6. Publication bias

As studies reporting statistically significant results may be more likely to be published than those reporting non-statistically significant ones (i.e. file drawer problem [91]), the mean effect size might be overestimated. To test for possible publication bias, we used funnel plots to represent the precision of each study (i.e. the inverse of the standard error) against the effect size value of the study (see electronic supplementary material, figure S2). When the data points are symmetrically distributed around the mean, this indicates an absence of publication bias. In addition to funnel plots, we performed an Egger's regression [92], which is a linear regression of the effect size value of each study against their precision. However, because the different effect sizes are not independent, we fitted the linear regression on the residuals of the meta-analysis (which are independent) against its precision [86]. In the absence of publication bias, the intercept of the regression should not differ from zero. Finally, we assessed whether our data were subjected to a time-lag bias (i.e. the decrease of the effect size over time) by including the publication year as a moderator in the meta-regression [93].

## 2.7. Outlier analysis

To avoid bias due to extreme and possibly spurious sex differences in TL, we checked for the presence of outliers. To identify outliers, we first calculated studentized deleted residuals, considering any study with studentized deleted residuals greater than 1.96 in absolute value as a potential outlier [94]. Then, the influence of such outliers was examined by calculating the DFFITS (a measure of how the predicted value change if an observation is excluded) and the Cook's distance (see [94] for more information). This diagnostic was done on the residuals of the meta-analysis using the package *metafor* [95]. Only one study, which focused on mandrills (*Mandrillux sphinx*) [68] was detected as a possible outlier ($g \pm 95\%$ CI = 1.28 ± 0.4, studentized deleted residuals = 4.05) (see electronic supplementary material, figure S3). All our analyses were therefore performed with and without the data from this species to assess its effects on the overall results.

## 2.8. The effect of sex differences in telomere length on sex differences in adult life expectancy

To assess whether sex differences in adult life expectancy could be explained by sex differences in TL, we performed a phylogenetically controlled comparative analysis. The adult life expectancy for both sexes of

**Table 2.** Number of studies, species and effect sizes included in the meta-analysis performed on juveniles and adults non-adjusted for age.

| | | |
|---|---|---|
| number of studies | total | 58 |
| | mammals | 13 |
| | birds | 33 |
| | reptiles | 6 |
| | fish | 6 |
| number of species | total | 52 |
| | mammals | 13 |
| | birds | 28 |
| | reptiles | 6 |
| | fish | 5 |
| number of effect sizes | total | 93 |
| | mammals | 31 |
| | birds | 38 |
| | reptiles | 15 |
| | fish | 9 |

each species was estimated using the formula: $s/(1-s)$ where $s$ is the mean annual adult survival [96]. Sex differences in adult life expectancy were then estimated as the log-transformed differences of the adult life expectancy between sexes. A Bayesian linear mixed model using MCMCglmm was built, with the sex difference in adult life expectancy as the dependent variable, the sex difference in TL (non-corrected for age with juveniles and adults) as the independent variable and with both female body mass and age at first reproduction as covariates. The random factors *phylogeny* and *species* as described for the meta-analysis model were also included in the model. Like for previous meta-analyses, models were run with non-informative priors (inverse Wishart prior with $v = 0.02$ and $V = 1$), with 1 000 000 iterations and the convergence was assessed by performing a Gelman–Rubin diagnostic [88]. Models were then ranked using the deviance information criterion (DIC) [97].

## 3. Results

The full dataset included 52 species that mostly corresponded to birds and mammals (55% and 23% of the species included, respectively, table 2).

To examine variation in sex differences in TL across vertebrates, we performed three separate meta-analyses (i.e. the first using data on juveniles and adults non-adjusted for age, the second using data on adults non-adjusted for age and the third using data on adults adjusted for age). Because the results were qualitatively similar for the three meta-analyses, we only present results from the first one, based on the largest dataset. Results from meta-analyses 2 and 3 are reported in electronic supplementary material, tables S3–S5.

The meta-analysis revealed no detectable sex differences in TL (mean = −0.038, HPDI = [−0.299: 0.233]) (figure 2). The random factors '*species*', '*population*' and '*sample*' accounted for a low (all specific $I^2$, the measure of the total variance explained by each random factor, values were smaller than or close to 0.25) but a similar proportion of heterogeneity observed among studies (table 3). We also found a low effect of the phylogeny with a $H^2$ (the percentage of the between-study variance explained by the phylogenetic effect) around 25% (table 3). The proportion of the total variance explained by the between-study variance ($I^2_{tot} = 0.71$, table 3) corresponded to a moderate heterogeneity value.

There was no detectable publication bias (see electronic supplementary material, figure S2) because data points displayed on the funnel plot, which represent studies, were symmetrically distributed around the mean. Moreover, the intercept of the Egger's regression did not differ from zero (mean ± s.e. = −0.001 ± 0.060). Effect sizes slightly increased over time (slope of the meta-regression = 0.018,

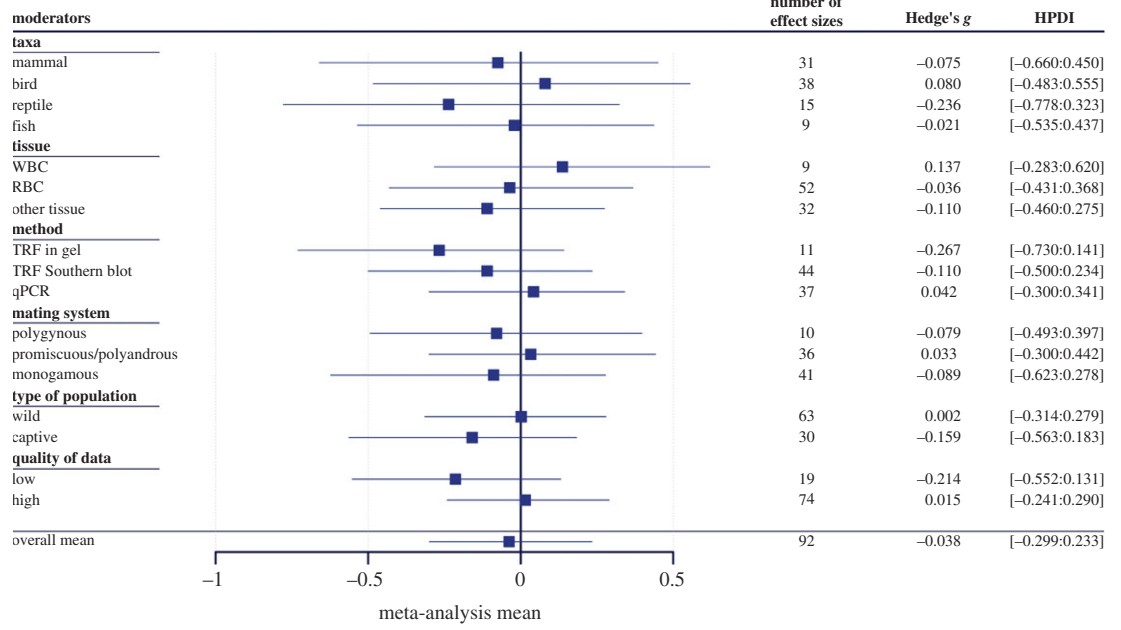

**Figure 2.** Mean of each moderator included in the meta-regression. All means are presented with their 95% highest posterior density intervals and sample size. A negative value means that females have longer telomeres than males while a positive value means that males have longer telomeres than females. Hedge's *g* corresponds to the value of the effect size and HPDI corresponds to the lower and upper high posterior density limits of the credible interval.

**Table 3.** *I*² values associated with each random effect (population, species and sample) and the phylogenetic heritability *H*² value, for the meta-analysis performed on juveniles and adults non-corrected for age. HPDI corresponds to the lower and upper high posterior density limits of the credible interval.

|  | mean | HPDI |
|---|---|---|
| $I^2$ population | 0.128 | [0.009 : 0.320] |
| $I^2$ species | 0.169 | [0.012 : 0.393] |
| $I^2$ sample | 0.134 | [0.006 : 0.340] |
| $I^2$ residuals | 0.083 | [0.010 : 0.225] |
| $H^2$ | 0.275 | [0.014 : 0.608] |
| $I^2$ total | 0.709 | [0.564 : 0.819] |

HPDI = [-0.003 : 0.038]), but this relationship was not statistically significant, providing no evidence for a time-lag bias (see electronic supplementary material, figure S4).

There was no evidence of sex differences in TL in any of the four vertebrate classes (figure 2). Sex differences in TL were not influenced by the method used to extract data from the original paper (mean difference in data extraction quality index: low quality versus high quality = −0.224, HPDI = [−0.479 : 0.039]), or whether the study populations were wild or captive (mean difference: 0.160, HPDI = [−0.138 : 0.455]). Likewise, neither the method used to measure TL nor the nature of the biological tissues had any detectable influence on sex differences in TL (figure 2). However, we found a small trend of longer TL in females when using the TRFI method, but this trend was not statistically significant (mean = −0.267, HPDI = [−0.730 : 0.141]). We did not detect any association between sex differences in TL and age at first reproduction (linear regression slope = 0.041, HPDI = [−0.068 : 0.148]).

An unexpected positive association between sex differences in TL and sexual size dimorphism was detected (slope = 0.522, HPDI = [0.043 : 0.967]), meaning that the larger the males are compared with the females, the longer their telomeres will be compared with females. However, when the study on mandrills, which was identified as an outlier, was removed from the analyses the intervals of the estimated effect of the sexual size dimorphism on the sex difference in TL included zero (slope = 0.212,

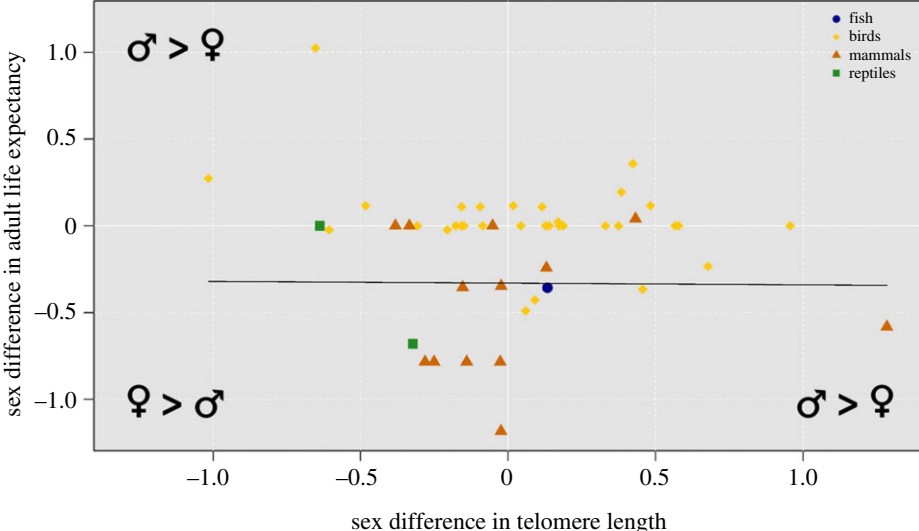

**Figure 3.** Absence of association between sex differences in adult life expectancy (log-transformed) and the effect size of the mean sex difference in TL for each species included in the analysis. The zero value corresponds to an absence of difference in adult life expectancy (y-axis) and in TL (x-axis). The black line represents the slope of the relationship between sex differences in adult life expectancy and sex differences in TL for the model that only includes the effect size of the mean sex difference in TL.

HPDI = [−0.233 : 0.674]). Sex differences in TL did not differ between mating systems (i.e. another proxy of the sex differences in life-history strategies, figure 2).

Sex differences in TL were not associated with sex differences in adult life expectancy. The model with the lowest DIC did not include sex differences in TL (slope of −0.009, credible interval 95% = [−0.092 : 0.072], $N = 36$ species, for the model that included only sex differences in TL, figure 3). Results were qualitatively unchanged when mammals and birds were analysed separately (see electronic supplementary material, table S6 for model selection tables).

## 4. Discussion

Once technical and biological confounding factors were taken into account, our meta-analysis revealed a lack of consistent sex differences in mean TL, both across vertebrates and within vertebrate classes. In addition, we did not find any sex differences in mean TL using age-corrected effect sizes (see electronic supplementary material, table S5), which suggest an absence of sex differences in telomere attrition patterns. Therefore, sex differences in mean TL that have been reported among several human populations [9] cannot be generalized to other mammals or other classes of vertebrates. Our results contrast with previous findings from a study that compiled 49 effect sizes and encompassed five species of mammals (mostly laboratory rodents and humans) [8], which suggested that, in mammals, telomeres are generally longer in females than in males. Our meta-analysis was performed on a wider range of mammalian populations in the wild (i.e. 12 species encompassing five orders) and does not support any consistent sex difference in TL in mammals. Thus, we found no support for the heterogametic sex hypothesis as an explanation for the sex differences in mean TL observed in humans [9]. In mammals, homogametic females had statistically longer telomeres than males in only 2 of 12 species, while in birds, homogametic males showed statistical evidence for longer telomere than females in only 4 of 28 species. It is therefore unlikely that differences in sex chromosome content explain the sex differences in TL across vertebrates. However, we cannot exclude that this lack of relationship is due to a taxonomic bias in the species included in our dataset, which favoured birds and mammals (40 species out of 52). These two vertebrate classes display a single genetic sex-determination system (i.e. ZZ/ZW and XX/XY, respectively). Adding data on reptile, fish and amphibian species (i.e. taxa that contains both XX/XY and ZZ/ZW sex-determination system), would allow disentangling the effect of the sex chromosome (heterogametic disadvantage hypothesis) from the effect of phylogeny.

Sex differences in life-history strategies are expected to be positively associated with sex differences in adult TL. More specifically, in species subjected to intense sexual selection, males allocate a substantial

amount of resources to sexual competition (e.g. growth and maintenance of costly secondary sexual traits) that might cause an increase in both the rate of cell division and the amount of oxidative damage [98] and ultimately lead to a faster rate of telomere attrition [99]. Contrary to our expectation, we found a positive association between the intensity of sexual size dimorphism and the amount of sex differences in telomere length. This result was largely driven by one study [68] on mandrill, a polygynous species where males are about three times bigger than females [100] and have longer telomeres (36% longer). However, this relationship was non-statistically significant when this species was removed from the analysis, suggesting that the sexual size dimorphism is unlikely to be universally linked with the sex difference in TL. Taken together, these results suggest that sexual selection has, at best, a limited influence on the evolution of sex differences in telomere length. Consistent with this conclusion, intraspecific studies to date testing the same hypothesis have revealed contrasting results. In the Australian painted dragon (*Ctenophorus pictus*) in which male coloration during the breeding season is tightly linked to the reproductive allocation and success, males that are better able to maintain their coloration also underwent more telomere erosion [101]. Conversely, in the common yellowthroat (*Geothlypis trichas*) males with higher UV brightness, a trait that is positively linked to reproductive success experienced a slower rate of telomere erosion [102]. Since sexual differentiation and many aspects of male and female physiology are driven by steroid sex hormones, it has been proposed that sex hormones could mediate sex differences in TL, notably by modulating the level of oxidative stress [8]. Accordingly, oestrogens would protect females against oxidative stress while testosterone would decrease resistance to oxidative stress in males [103]. As the level of testosterone increases with the intensity of sexual size dimorphism and with the expression of costly male secondary sexual traits [103], our results indirectly challenge the belief that sex hormones play a key role in mediating sex differences in TL in humans.

The discrepancy in terms of sex-specific TL between humans and other vertebrate species may in part be relative to the exceptionally high longevity of humans. Since human TL is generally measured in the elderly (mean age in [9] ranging from 37 to 89.9 years old with an overall mean of 55.8 years old), the reported sex differences in TL might be particularly pronounced at old ages (although we note that sex differences were also found in umbilical cord blood, [10]). By contrast, most individuals measured in vertebrate populations in the wild are juveniles and young adults. Thus, sex differences in TL may not be observed in these studies if they occur in late adulthood. This hypothesis remains to be tested by measuring sex-specific TL in old individuals of other very long-lived vertebrate species such as naked mole rats, elephants, bats, seabirds or turtles. In addition, due to cultural habits, human TL is more prone to suffer from lifestyle and environmental factors than that of animals. Factors such as stress, diet, physical activity and harmful consumption are known to influence telomere length (see [104] for a review), possibly in a sex-specific way. However, results about lifestyle and TL are contrasted (see [105] where no effect of smoking was found on telomere attrition) and no consensus has been reached yet.

The comparative analysis we performed indicates that sex differences in adult life expectancy are not related to sex differences in telomere length across our set of vertebrate species. This finding matches the results of a recent meta-analysis performed across 20 vertebrate species, which reported that there is an association between telomere length and mortality risk but that this association is not sex-specific [6]. Our starting hypothesis was based on the existence of a causal relationship between TL and ageing, which remains debated [3]. A lack of causal association between TL and ageing would obviously explain the absence of sex differences in telomere length, as well as the lack of association between sex differences in telomere length and sex differences in lifespan we report. Taken together, these results suggest that telomere dynamics do not provide an explanation for the sex differences in longevity observed across vertebrates. On the other hand, if TL and ageing are causally linked, our findings might come from a lack of power of our dataset. Note, however, that under that scenario it would be quite difficult to explain the trend for a positive association between longer relative TL in males and intensity of sexual size dimorphism.

The heterogeneity analysis revealed that the proportion of the total variance explained by the between-study variance was quite low compared with that generally reported in meta-analyses in the field of evolutionary biology. Indeed, the average heterogeneity value computed in evolutionary meta-analyses is around 92% because these analyses are not confined to a single species like meta-analyses commonly performed in medical sciences, which probably increases heterogeneity [106]. Hence, our measure of the total variance explained by the between-study variance ($I^2 = 0.71$) shows that heterogeneity within studies (i.e. the sampling variances associated with each effect size) accounts for a substantial proportion of the total variance in TL. This result could be due to a large number of studies with a small sample size that were included in our meta-analysis. Indeed, 30% of the studies

in our meta-analysis had a sample size less than or equal to 20 individuals, which could impair our capacity to detect effects from variables included as moderators (i.e. the meta-analyses could be underpowered if the overall effect is expected to be small [93]). Interestingly, sex differences reported in humans are also weak in magnitude but generally based on much larger datasets (e.g. over 100 000 participants for the biggest study to date, [107]), which results in statistically significant sex differences. Therefore, if there are consistent but very weak differences between sexes in mammals, we might have not enough power to detect them. However, when our analysis was performed on effect sizes calculated from studies with more than 40 individuals, our measure of heterogeneity increased ($I^2 = 0.84$) (see electronic supplementary material, table S1), but results were qualitatively unchanged, suggesting that the overall absence of sex differences we found was not due to a power issue. However, it is worth noticing that for some moderators (e.g. only 9 effect sizes measured in white blood cells, only 11 effect sizes measured using TRFI), the number of computed effect sizes was very low, which might have prevented us to detect an effect of the tissue or the method in the meta-regression.

Our results were robust to publication bias, which contrasts with a previous meta-analysis looking at the relationship between TL and mortality in non-model vertebrates [6] where a substantial publication bias was reported. The absence of a publication bias in our analyses could be explained by the fact that sex-specific TL measurements we compiled for the meta-analysis were generally not the main focus of the papers we retrieved.

We failed to find any consistent sex differences in TL across vertebrates or within any single class of vertebrates. This absence of a difference does not support the main evolutionary hypotheses (i.e. the heterogametic sex disadvantage and the life-history hypotheses) that have been proposed to explain sex differences in TL in humans. Our findings also call into question the possible role played by sex hormones in sex differences in TL. Testing hypotheses across broad sets of species beyond humans is essential to understand the evolutionary roots of sex differences in TL. The increasing number of publications on many different species should allow clarifying the causes and consequences of TL in the next future and should be beneficial for both evolutionary biology and medical sciences.

Data accessibility. Data used in the meta-analyses are published as the electronic supplementary material.

Authors' contributions. F.R., D.H.N. and J.-F.L. conceived the study with inputs from H.F., B.R. and J.-M.G. F.R compiled the dataset. F.R. and V.R. analysed the data with inputs from J.-F.L. and J.-M.G. F.R. wrote the first draft of the paper and then received input from all authors.

Competing interests. The authors declare no competing interests.

Funding. F.R. is funded by the French Ministry of Education and Research. J.-F.L. B.R. and J.-M.G. are supported by a grant from the Agence Nationale de la Recherche (grant no. ANR-15-CE32-0002-01 to J.-F.L.). This work is supported by the Charity Open Access Fund and the Royal Society international exchange grant: Telomere dynamics in wild populations of vertebrates (grant no. IEC\R2\181087). This study was performed within the framework of the LABEX ECOFECT (grant no. ANR-11-LABX-0048) of Université de Lyon, within the programme 'Investissements d'Avenir' (grant no. ANR-11-IDEX-0007) operated by the French National Research Agency (ANR).

Acknowledgements. All authors would like to thank the following for providing data, in alphabetical order: Pierre Bize, Donald Blomqvist, Andréaz Dupoué, Britt Heidinger, Patrik Karell, Yuichi Mizutani, Pat Monaghan, Jenny Ouyang, Pablo Salmon, Simon Verhulst and Rebecca Young. We also want to thank Jack Cerchiara, Alessandra Costanzo, Mark Haussmann, Sin-Yeon Kim, Nuria Mach, Marco Parolini, Sophie Reichert and Oscar Vedder for providing data or information we were not able to include in the meta-analysis. We are grateful to two anonymous reviewers for their comments that markedly improved our manuscript. We acknowledge BBSRC funding to D.H.N.

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
