## [Reviewer comments · Royal Society Open Science]

Review History

RSOS-200548.R0 (Original submission)

Review form: Reviewer 1

Is the manuscript scientifically sound in its present form?

Yes

Are the interpretations and conclusions justified by the results?

Yes

Is the language acceptable?

Yes

Do you have any ethical concerns with this paper?

No

Have you any concerns about statistical analyses in this paper?

Yes

Recommendation?

Accept with minor revision (please list in comments)

Comments to the Author(s)

The manuscript entitled “No sex differences in adult telomere length across vertebrates: A meta-analysis” by Remot and colleagues investigate if the sex differences in lifespan seen across vertebrates can be explained by sex differences in telomere length by analyzing publicly available data from 51 taxa. They find no support for the starting hypothesis, and also don’t find support for the heterogametic sex advantage hypothesis or the life history hypothesis as explanations for sex differences in telomere length.

The manuscript is a revised manuscript that has been evaluated previously for publication in the Royal Society Proceedings B and upon rejection was transferred to the Royal Society’s open access journal Royal Society Open Science. The authors have provided a detailed response to reviewers and have made significant efforts to improve the manuscript.

General comments:

- The results section is difficult to read and would benefit from additional editing and inclusion of some of the basic information necessary to understand the analyses carried out by the authors. As is, the results section is only understandable if the reader is willing to dig through the methods section for details.
- The question of how telomere length might be related to sex differences in aging is interesting and relevant. However, available data from multiple taxa indicate that telomere dynamics are complex, even within species, with telomere attrition patterns depending on the tissue examined, being impacted by stress, and non-linear throughout the lifespan. In addition, there are significant differences in the sensitivities of the methods typically used to assay telomere length. Given these facts, I am not surprised by the authors inability to find a consistent pattern in this meta-analysis. However, I’m also not convinced that this is due to the lack of a relationship between telomere attrition, aging, and sex, and not a limitation of the data available. I wonder if the authors could expand their discussion to address this point and to make some recommendations as to what data we would need to properly address the questions posed.
- The phylogenetic distribution of the species included in the study is rather biased and limited as the authors acknowledge. I wonder how these biases and limitations impact the authors findings, and how a more equal sampling across taxa would change the results.
- Generally speaking, I think this manuscript would benefit from a more in-depth discussion of the limitations of the study and the inherent limitations of the available data.

Review form: Reviewer 2

Is the manuscript scientifically sound in its present form?

Yes

Are the interpretations and conclusions justified by the results?

Yes

Is the language acceptable?

Yes

Do you have any ethical concerns with this paper?

No

Have you any concerns about statistical analyses in this paper?

Yes

Recommendation?

Accept with minor revision (please list in comments)

Comments to the Author(s)

This is a well conducted meta-analysis on an important topic. There are some issues below, that I am sure the authors can resolve, but do require a great deal of care, and some considerable revision. I reviewed this independently without looking at the earlier reviews to give the author's an objective review. Happy to look at another version, but I am confident with this guidance the authors can make the changes required.

Specific comments:

L38-42 This under the assumption that TL and ageing are causally related which is an assumption rather than proof in my opinion. One other explanation for your null findings is the last, TL does not explain ageing or is not a great biomarker.

L55. I would argue this is false. The alternative TL is a biomarker and not causal needs to be discussed. You acknowledge this only briefly.

L65-66. Is this correcting for their difference in karyotype, i.e. two X in female? If X has relatively longer TL, gel and qPCR methods will detect longer overall TL. Line 73. Again karyotype is obvious explanation here? Ah I see you get to this now at L76.

I guess it needs to be clear that the karyote explanation is not one of causal involvement of telomeres in the sex difference I guess. Or do you think males would die because they have short telomeres on the Y or something. I doubt it.

L86-87. Telomere shortening on which chromosome, you seem to write about some meta overarching telomere shortening or telomeres, whereas they of course actually are chromosome specific, and you should relate this accordingly to the karyote hypothesis. Also I would make clear this is just a telomere specific explanation of sex differences in lifespan anyways.

L99. Not ref 11 I assume?

L99-L103. You need to make clear TL is maybe a way in to understand ageing, but you cannot equate it to ageing, you seem to suggest here that you can.

L108-109. Not representative how?

L111. Unclear if you now will investigate dynamics of or absolute TL.

L117. The question is do you have enough variation in karyotype switching between male and female amongst relatively closely related species to tease that apart from phylogeny?

L164. Don't you expect unequal variances between the sexes?

L186. Why is overfitting a problem here? As they are so correlated to each other? I would like to see how strongly and a full model none the less as overfitting is actually quite hard to achieve if you have >10 times the datapoints as independents which seems to be the case here?

L194. I do not think the 'website' did this.

L200. Unclear what 'population' is. Unclear what 'sample' is if you do not have multiple effect sizes per study this is not needed.

L201-202. This is a partially flawed interpretation of random effects. You will correct for their dependency but not for their precision. So you might double count individuals so you will still bias your results. You are better off adjusting mev for this reduction in confidence, i.e. using some kind of independent sample size to calculate that. Or you could just include the largest sample size per population studied? I think earlier in the methods you said you did?

Line 202. Isnt your random effect structure missing 'species'? This is different than including the relatedness matrix.

Line 210. What is you gelman-rubin convergence cut-off value?

Line 218-221. This is unconventional in my opinion. The mere use of meta-analysis is that it gives you an overall estimate from estimates that are of known sampling variance (determined by N). Did you assess publication bias?

L226. For many species it will be impossible to distinguish juveniles from adults I would assume?

L237-240. I think you mentioning dynamics quite often in the introduction put me on the wrong foot that you were actually looking at differential shortening rate between the sexes, but you are not. I would make this more explicit.

Outlier. I am not convinced this adds a lot really, maybe to try and prove the null hypothesis (which is a fallacy as well), also a bit strange you rely on metafor her to provide you the residual and not MCMCglmm.

L310. In this model correcting for the one most explanatory variable in your model (gel versus qPCR) what was the estimate of the sex effect at the intercept? Positive, if also negative you might have a sex effect within gel only. Not that it matters much but worth reporting.

L315, rephrase 'will be'.

L315. This effect cannot really be appreciated without knowing what the intercept is in this model. Needs presenting and interpreting.

L317. I am not very convinced by this outlier analysis. Point of meta-analysis is to quantitatively summarise all data available. Yes worth mentioning this hinges on the extremes and driven by one datapoint (partially). I do not think it is proof this relationship is driven by the outlier or that this point is an actual outlier. What is it an outlier in? In biology? In the sampling variance/error distribution? You won't be able to distinguish those options.

L319. Shouldn't sexual dimorphism in size be logged?

L331. How does the data compare in terms of effect size? Do you detect a similar effect size, but your precision is just lower using a lower overall sample size than in human studies? Are these human studies age-corrected?

L338-340. I do not see this result or test of this in your result section?

L351-353. The relationship did not disappear, if you test the resulting slope against each other with and without the outlier that won't be different I don't think. I do not think your interpretation of outlier analysis is correct.

L369-371. You will need to review the ages of the human studies that claim sex differences for this. How does this fit with the relationship between TL and mortality being lower in older humans (or completely gone in old people..).

L393-398. This is wrong. Sample size included should not influence I^2 .

L402-403. You want to compare effect size. I am unclear as to why you are not formally comparing that to the human effect. It might actually not differ from that at all. You have a testable hypothesis here and a meta-analysis to quantitatively compare effects and their confidence/credibility intervals.

L405-407. Again sample size included should not influence I^2 . I think you misunderstand this part of the meta-analytic methodology?

I think your discussion needs an appreciation of telomere attrition over absolute telomere length differences and what they tell us about the biology of ageing. For example recent comparative analyses show telomere shortening related to a species' lifespan. Yet telomere shortening is less predictive of mortality in humans than absolute telomere length is. For other vertebrates I would not know what the current consensus is.

Decision letter (RSOS-200548.R0)

Dear Mr Remot,

On behalf of the Editors, I am pleased to inform you that your Manuscript RSOS-200548 entitled "No sex differences in adult telomere length across vertebrates: A meta-analysis" has been accepted for publication in Royal Society Open Science subject to minor revision in accordance with the referee suggestions. Please find the referees' comments at the end of this email.

The reviewers and handling editors have recommended publication, but also suggest some minor revisions to your manuscript. Therefore, I invite you to respond to the comments and revise your manuscript.

- Ethics statement

- Data accessibility

<http://datadryad.org/submit?journalID=RSOS&manu=RSOS-200548>

- Competing interests

- Authors' contributions

- Acknowledgements

- Funding statement

Because the schedule for publication is very tight, it is a condition of publication that you submit the revised version of your manuscript before 05-Aug-2020. Please note that the revision

deadline will expire at 00.00am on this date. If you do not think you will be able to meet this date please let me know immediately.

If your manuscript is newly submitted and subsequently accepted for publication, you will be asked to pay the article processing charge, unless you request a waiver and this is approved by Royal Society Publishing. You can find out more about the charges at <https://royalsocietypublishing.org/rsos/charges>. Should you have any queries, please contact opscience@royalsociety.org.

on behalf of the Associate Editor, and Professor Kevin Padian (Subject Editor)
openscience@royalsociety.org

Associate Editor Comments to Author:

Two reviewers have now assessed your transferred manuscript and are broadly in favour of publication, but please do ensure you carefully address the queries and comments the referees make and include a full point-by-point response when you submit your revision.

Reviewer comments to Author:

Reviewer: 1
Comments to the Author(s)

The manuscript entitled "No sex differences in adult telomere length across vertebrates: A meta-analysis" by Remot and colleagues investigate if the sex differences in lifespan seen across vertebrates can be explained by sex differences in telomere length by analyzing publicly available data from 51 taxa. They find no support for the starting hypothesis, and also don't find support for the heterogametic sex advantage hypothesis or the life history hypothesis as explanations for sex differences in telomere length.

The manuscript is a revised manuscript that has been evaluated previously for publication in the Royal Society Proceedings B and upon rejection was transferred to the Royal Society's open access journal Royal Society Open Science. The authors have provided a detailed response to reviewers and have made significant efforts to improve the manuscript.

General comments:

- The results section is difficult to read and would benefit from additional editing and inclusion of some of the basic information necessary to understand the analyses carried out by the authors. As is, the results section is only understandable if the reader is willing to dig through the methods section for details.
- The question of how telomere length might be related to sex differences in aging is interesting and relevant. However, available data from multiple taxa indicate that telomere dynamics are complex, even within species, with telomere attrition patterns depending on the tissue examined, being impacted by stress, and non-linear throughout the lifespan. In addition, there are significant differences in the sensitivities of the methods typically used to assay telomere length. Given these facts, I am not surprised by the authors inability to find a consistent pattern in this meta-analysis. However, I'm also not convinced that this is due to the lack of a relationship between telomere attrition, aging, and sex, and not a limitation of the data available. I wonder if the authors could expand their discussion to address this point and to make some recommendations as to what data we would need to properly address the questions posed.

- The phylogenetic distribution of the species included in the study is rather biased and limited as the authors acknowledge. I wonder how these biases and limitations impact the authors findings, and how a more equal sampling across taxa would change the results.
- Generally speaking, I think this manuscript would benefit from a more in-depth discussion of the limitations of the study and the inherent limitations of the available data.

Reviewer: 2

Comments to the Author(s)

This is a well conducted meta-analysis on an important topic. There are some issues below, that I am sure the authors can resolve, but do require a great deal of care, and some considerable revision. I reviewed this independently without looking at the earlier reviews to give the author's an objective review. Happy to look at another version, but I am confident with this guidance the authors can make the changes required.

Specific comments:

L38-42 This under the assumption that TL and ageing are causally related which is an assumption rather than proof in my opinion. One other explanation for your null findings is the last, TL does not explain ageing or is not a great biomarker.

L55. I would argue this is false. The alternative TL is a biomarker and not causal needs to be discussed. You acknowledge this only briefly.

L65-66. Is this correcting for their difference in karyotype, i.e. two X in female? If X has relatively longer TL, gel and qPCR methods will detect longer overall TL. Line 73. Again karyotype is obvious explanation here? Ah I see you get to this now at L76.

I guess it needs to be clear that the karyote explanation is not one of causal involvement of telomeres in the sex difference I guess. Or do you think males would die because they have short telomeres on the Y or something. I doubt it.

L86-87. Telomere shortening on which chromosome, you seem to write about some meta overarching telomere shortening or telomeres, whereas they of course actually are chromosome specific, and you should relate this accordingly to the karyote hypothesis. Also I would make clear this is just a telomere specific explanation of sex differences in lifespan anyways.

L99. Not ref 11 I assume?

L99-L103. You need to make clear TL is maybe a way in to understand ageing, but you cannot equate it to ageing, you seem to suggest here that you can.

L108-109. Not representative how?

L111. Unclear if you now will investigate dynamics of or absolute TL.

L117. The question is do you have enough variation in karyotype switching between male and female amongst relatively closely related species to tease that apart from phylogeny?

L164. Don't you expect unequal variances between the sexes?

L186. Why is overfitting a problem here? As they are so correlated to each other? I would like to see how strongly and a full model none the less as overfitting is actually quite hard to achieve if you have >10 times the datapoints as independents which seems to be the case here?

L194. I do not think the 'website' did this.

L200. Unclear what 'population' is. Unclear what 'sample' is if you do not have multiple effect sizes per study this is not needed.

L201-202. This is a partially flawed interpretation of random effects. You will correct for their dependency but not for their precision. So you might double count individuals so you will still bias your results. You are better off adjusting mev for this reduction in confidence, i.e. using some kind of independent sample size to calculate that. Or you could just include the largest sample size per population studied? I think earlier in the methods you said you did?

Line 202. Isnt your random effect structure missing 'species'? This is different than including the relatedness matrix.

Line 210. What is your gelman-rubin convergence cut-off value?

Line 218-221. This is unconventional in my opinion. The mere use of meta-analysis is that it gives you an overall estimate from estimates that are of known sampling variance (determined by N).

Did you assess publication bias?

L226. For many species it will be impossible to distinguish juveniles from adults I would assume?

L237-240. I think you mentioning dynamics quite often in the introduction put me on the wrong foot that you were actually looking at differential shortening rate between the sexes, but you are not. I would make this more explicit.

Outlier. I am not convinced this adds a lot really, maybe to try and prove the null hypothesis (which is a fallacy as well), also a bit strange you rely on metafor her to provide you the residual and not MCMCglmm.

L310. In this model correcting for the one most explanatory variable in your model (gel versus qPCR) what was the estimate of the sex effect at the intercept? Positive, if also negative you might have a sex effect within gel only. Not that it matters much but worth reporting.

L315, rephrase 'will be'.

L315. This effect cannot really be appreciated without knowing what the intercept is in this model. Needs presenting and interpreting.

L317. I am not very convinced by this outlier analysis. Point of meta-analysis is to quantitatively summarise all data available. Yes worth mentioning this hinges on the extremes and driven by one datapoint (partially). I do not think it is proof this relationship is driven by the outlier or that this point is an actual outlier. What is it an outlier in? In biology? In the sampling variance/error distribution? You won't be able to distinguish those options.

L319. Shouldn't sexual dimorphism in size be logged?

L331. How does the data compare in terms of effect size? Do you detect a similar effect size, but your precision is just lower using a lower overall sample size than in human studies? Are these human studies age-corrected?

L338-340. I do not see this result or test of this in your result section?

L351-353. The relationship did not disappear, if you test the resulting slope against each other with and without the outlier that wont be different I don't think. I do not think your interpretation of outlier analysis is correct.

L369-371. You will need to review the ages of the human studies that claim sex differences for this. How does this fit with the relationship between TL and mortality being lower in older humans (or completely gone in old people..).

L393-398. This is wrong. Sample size included should not influence I^2 .

L402-403. You want to compare effect size. I am unclear as to why you are not formally comparing that to the human effect. It might actually not differ from that at all. You have a testable hypothesis here and a meta-analysis to quantitatively compare effects and their confidence/credibility intervals.

L405-407. Again sample size included should not influence I^2 . I think you misunderstand this part of the meta-analytic methodology?

I think your discussion needs an appreciation of telomere attrition over absolute telomere length differences and what they tell us about the biology of ageing. For example recent comparative analyses show telomere shortening related to a species' lifespan. Yet telomere shortening is less predictive of mortality in humans than absolute telomere length is. For other vertebrates I would not know what the current consensus is.

Author's Response to Decision Letter for (RSOS-200548.R0)

See Appendix A.

Decision letter (RSOS-200548.R1)

Dear Mr Remot,

It is a pleasure to accept your manuscript entitled "No sex differences in adult telomere length across vertebrates: A meta-analysis" in its current form for publication in Royal Society Open Science.

on behalf of Prof Kevin Padian (Subject Editor)
openscience@royalsociety.org

Appendix A

Dear Mr Remot,

On behalf of the Editors, I am pleased to inform you that your Manuscript RSOS-200548 entitled "No sex differences in adult telomere length across vertebrates: A meta-analysis" has been accepted for publication in Royal Society Open Science subject to minor revision in accordance with the referee suggestions. Please find the referees' comments at the end of this email.

Associate Editor Comments to Author:

Two reviewers have now assessed your transferred manuscript and are broadly in favour of publication, but please do ensure you carefully address the queries and comments the referees make and include a full point-by-point response when you submit your revision.

Dear Editor,

We thank the two reviewers and the Associate Editor for their positive evaluation of our work and for their very constructive comments and suggestions. In our revised manuscript, we have addressed in depth each of the issues raised by the two reviewers. We also provide a detailed response to each comment (in bold).

Reviewer comments to Author:

Reviewer: 1

Comments to the Author(s)

The manuscript entitled "No sex differences in adult telomere length across vertebrates: A meta-analysis" by Remot and colleagues investigate if the sex differences in lifespan seen across vertebrates can be explained by sex differences in telomere length by analyzing publicly available data from 51 taxa. They find no support for the starting hypothesis, and also don't find support for the heterogametic sex advantage hypothesis or the life history hypothesis as explanations for sex differences in telomere length.

The manuscript is a revised manuscript that has been evaluated previously for publication in the Royal Society Proceedings B and upon rejection was transferred to the Royal Society's open access journal Royal Society Open Science. The authors have provided a detailed response to reviewers and have made significant efforts to improve the manuscript.

Our response: We warmly thank the referee for her/his positive and thorough evaluation of our work.

General comments:

- The results section is difficult to read and would benefit from additional editing and inclusion of some of the basic information necessary to understand the analyses carried out by the authors. As is, the results section is only understandable if the reader is willing to dig through the methods section for details.

Our response: The referee is not very specific about what information is missing from the result section, so that it is difficult to identify precisely what would require additional information. However, according to this comment, we have made sure that all the necessary information regarding the analyses is included in the Material and Methods section of the revised version.

- The question of how telomere length might be related to sex differences in aging is interesting and relevant. However, available data from multiple taxa indicate that telomere dynamics are complex, even within species, with telomere attrition patterns depending on the tissue examined, being impacted by stress, and non-linear throughout the lifespan. In addition, there are significant differences in the sensitivities of the methods typically used to assay telomere length. Given these facts, I am not surprised by the authors inability to find a consistent pattern in this meta-analysis. However, I'm also not convinced that this is due to the lack of a relationship between telomere attrition, aging, and sex, and not a limitation of the data available. I wonder if the authors could expand their discussion to address this point and to make some recommendations as to what data we would need to properly address the questions posed.

Our response: As rightly stated by the referee, the absence of sex differences in telomere length can be interpreted in two ways. The first is that there are no overall sex differences in telomere length in vertebrates. The second, is that the quality of the data we gathered does not allow us to detect any existing effect. To improve our ability to disentangle these scenarios, we controlled for various sources of heterogeneity such as the telomere measurement method and the type of tissue sampled, and included random effects to account for the non-independence of the measurements. However, we still agree with the referee that we might have lacked statistical power for some factors (*e.g.* only 9 effect sizes measured in white blood cells, only 11 effect sizes measured using TRFI). We now explicitly address this limitation of our study in the discussion section of the revised manuscript (lines 440-444).

- The phylogenetic distribution of the species included in the study is rather biased and limited as the authors acknowledge. I wonder how these biases and limitations impact the authors findings, and how a more equal sampling across taxa would change the results.

Our response: There is no reason to expect sex differences in telomere length in fish and reptiles to be different from those in birds and mammals. However, more data on reptiles, fish and amphibians would allow testing the heterogametic sex disadvantage hypothesis much more accurately because these groups display variation in sex determination system in contrary to birds and mammals, which would make possible to disentangle the effect of sex chromosome from the effect of phylogeny. This is now mentioned in the revised version of the manuscript (lines 358-364).

- Generally speaking, I think this manuscript would benefit from a more in-depth discussion of the limitations of the study and the inherent limitations of the available data.

Our response: Please see our answer to the previous comments.

Reviewer: 2

Comments to the Author(s)

This is a well conducted meta-analysis on an important topic. There are some issues below, that I am sure the authors can resolve, but do require a great deal of care, and some considerable revision. I reviewed this independently without looking at the earlier reviews to give the author's an objective review. Happy to look at another version, but I am confident with this guidance the authors can make the changes required.

Our response: We thank the referee for her/his positive evaluation of our work and her/his thoughtful comments.

Specific comments:

L38-42 This under the assumption that TL and ageing are causally related which is an assumption rather than proof in my opinion. One other explanation for your null findings is the last, TL does not explain ageing or is not a great biomarker.

Our response: Our hypothesis that sex differences in telomere length should match sex differences in lifespan/ageing across vertebrates is indeed based on the assumption that telomere dynamics is causally related to ageing. While there is now some clear evidence in the literature that short telomeres or steep rate of telomere attrition can impair lifespan ([1–3] in humans, laboratory organisms and populations in the wild), causal relationships are indeed difficult to establish (but see [4] for a case-study in wild-type mice). That is why we were very cautious in our wording (e.g. “While a causal association among telomere length (TL), age-specific diseases, and survival prospects remains uncertain” (lines 56-57)). Overall, we agree with the referee that we cannot ruled out that the absence of causal relationship might explain why we did not detect any sex difference in telomere length or an absence of relationship between sex differences in telomere length and sex differences in lifespan. This is now explicitly mentioned in the discussion of the revised manuscript (lines 411-415).

L55. I would argue this is false. The alternative TL is a biomarker and not causal needs to be discussed. You acknowledge this only briefly.

Our response: In the introduction, we remained very cautious regarding the causal relationship between telomere length and ageing ‘While a causal association among telomere length (TL), age-specific diseases, and survival prospects remains uncertain’ (see lines 56-57). However, following the reviewer’s concern, we added a discussion on how an absence of causal relationship could explain our results (see lines 411-415 and our response to the previous comment).

L65-66. Is this correcting for their difference in karyotype, i.e. two X in female? If X has relatively longer TL, gel and qPCR methods will detect longer overall TL. Line 73. Again

karyotype is obvious explanation here? Ah I see you get to this now at L76. I guess it needs to be clear that the karyote explanation is not one of causal involvement of telomeres in the sex difference I guess. Or do you think males would die because they have short telomeres on the Y or something. I doubt it.

Our response: Regarding the two studies we quoted (i.e. [5] and [6]), the authors did not correct for the difference in karyotype between males and females. It is important to mention here that most studies that measured telomere length have used either the TRF method or the qPCR method, which do not measure chromosome-specific telomere length but rather an average telomere length on a set of chromosomes in a sample of cells. However, we fully agree with the referee that differences in sex chromosome content are not supposed to explain sex differences in telomere length. Indeed, although sex differences have been found on the q-arms of the X chromosome (estimated to be 1100 bp shorter in boys than in girls shortly after birth [7]), this difference is present in only one pair of chromosomes out of the 24 pairs of chromosomes of humans and should contribute only little to the overall sex differences in telomere length. Here, our hypothesis that sex differences in sex chromosomes should be associated with sex differences in telomere length relies on the presence of genes related to telomere maintenance (e.g. DKC1) on the X chromosome in humans, as initially proposed by Barrett and Richardson [8]. We have reworded the text to make this point clearer in the revised manuscript (lines 83-86).

L86-87. Telomere shortening on which chromosome, you seem to write about some meta overarching telomere shortening or telomeres, whereas they of course actually are chromosome specific, and you should relate this accordingly to the karyote hypothesis. Also I would make clear this is just a telomere specific explanation of sex differences in lifespan anyways.

Our response: As explained in our response to the previous comment, we do not expect a specific effect on sex chromosome (e.g. shorter telomere length in Y) but on the average length of telomeres in a sample of cells. Our hypothesis (following Barrett & Richardson 2011, [8]) relies on the presence of a telomere maintenance allele that is located on the X chromosome but does not occur on the Y chromosome. Assuming that these genes are always present on the homogametic sex chromosome, the opposite relationship should be found in birds (i.e. shorter telomere in females than in males).

L99. Not ref 11 I assume?

Our response: We agree with the referee that a few comparative studies (including or focusing on animals in the wild) have also studied the genetic/physiological basis of ageing such as Xirocostas et al. 2020 [9] (ref 11 in our manuscript). This is now acknowledged in our revised version (line 104-105).

L99-L103. You need to make clear TL is maybe a way in to understand ageing, but you cannot equate it to ageing, you seem to suggest here that you can.

Our response: There was clearly a misunderstanding here, which was caused by the lack of clarity of the initial version of our work. We made clear in the revised version that sex-

specific telomere length might contribute to sex differences in health and lifespan, among other factors. This is now specified in our revised manuscript (lines 106-107).

L108-109. Not representative how?

Our response: Telomere length is known to be particularly long in laboratory mice (ranging from 30 to 200kb [10]) in comparison to those of wild-derived mice (< 25kb [10]). This is now mentioned in the revised manuscript (lines 116-117).

L111. Unclear if you now will investigate dynamics of or absolute TL.

Our response: We apologize for the lack of clarity in our initial version. We have now specified in the revised version that we were interested in analysing sex differences in mean telomere length to avoid any confusion (lines 121 and 123).

L117. The question is do you have enough variation in karyotype switching between male and female amongst relatively closely related species to tease that apart from phylogeny?

Our response: Since birds and mammals have only one sex determination system (i.e. XX in mammals and ZW in birds), testing the heterogametic sex disadvantage hypothesis in those species is challenging because the effect is confounded with that of phylogeny. By increasing the number of reptile, fish and amphibian species we could disentangle these effects, as each of these classes display a diversity of sex determination system (see also above our response to reviewer 1). We have now discussed this point in the revised manuscript (lines 358-364).

L164. Don't you expect unequal variances between the sexes?

Our response: Unequal variances in telomere length could occur between the sexes. However, the metric of effect size we used, the Hedge's g , is unaffected by unequal sampling variance (See [11]). This is now specified in lines 173 of the revised manuscript.

L186. Why is overfitting a problem here? As they are so correlated to each other? I would like to see how strongly and a full model none the less as overfitting is actually quite hard to achieve if you have >10 times the datapoints as independents which seems to be the case here?

Our response: In the meta-regression we tested 8 moderators (including 6 qualitative variables and 2 quantitative variables). This means that a full model would have required to estimate 18 parameters and 4 random factors, leading to 93 measures of effect sizes. Moreover, due to a high collinearity between body mass and age at first reproduction across species ($r = 0.71$, higher than the threshold of 0.7 advocated by Dormann et al. [12]), we analysed separately these moderators. This is why we decided to perform subset analyses rather than doing a single model selection procedure starting with the most complete model.

L194. I do not think the 'website' did this.

Our response: This has now been corrected in the revised manuscript (lines 202-203).

L200. Unclear what ‘population’ is. Unclear what ‘sample’ is if you do not have multiple effect sizes per study this is not needed.

Our response: The random factor ‘population’ refers to the case where multiple populations were sampled within a species (e.g. two populations for roe deer (*Capreolus capreolus*) [13]). The random factor ‘sample’ account for the fact that different effect sizes were measured from the same individuals using different methods or involving different tissues (e.g. 12 different tissues measured in the same individuals in the Crab-eating macaque (*Macaca fascicularis*)[14]). Therefore, the value of sample (ranging from 1 to 72) characterizes the set of individuals from which telomeres were measured, meaning that effect sizes that are calculated from the same individuals have the same ‘sample’ value. We have reworded these sentences to make these points clearer in the revised manuscript (see lines 210-215).

L201-202. This is a partially flawed interpretation of random effects. You will correct for their dependency but not for their precision. So you might double count individuals so you will still bias your results. You are better off adjusting mev for this reduction in confidence, i.e. using some kind of independent sample size to calculate that. Or you could just include the largest sample size per population studied? I think earlier in the methods you said you did?

Our response: There is a misunderstanding here, which was likely caused by the lack of clarity in the initial version of our work. As explained in our response to the previous referee’s comment, the random factor ‘sample’ accounts for the fact that different effect sizes have been computed from the same set of individuals. However, if we had decided to limit our analysis to the largest sample size per population, we would have reduced drastically the number of effect sizes (i.e. by 33%). Finally, the sentence in the methods that the reviewer is referring to (lines 156-159) is about the use of the same dataset (partially or totally) for different articles. We have reworded the text to make this point clearer in the revised manuscript (lines 156-159).

Line 202. Isn't your random effect structure missing ‘species’? This is different than including the relatedness matrix.

Our response: We apologize for the lack of clarity in our original manuscript. The random factor ‘species’ is indeed different from the random factor ‘phylogeny’. The random factor ‘phylogeny’ aims to control for phylogenetic relatedness among species. On the contrary, the random factor ‘species’ is independent of the phylogeny and takes into account that one species can correspond to multiple effect sizes in our dataset. We clarified this statement in our revised manuscript (line 213-215).

Line 210. What is you gelman-rubin convergence cut-off value?

Our response: According to Brooks & Gelman [15], for assessing convergence, we used a threshold value of 1.1. The values of convergence we obtained were close to 1.01 each time. We have now made clear what was the cut-off value in the revised version of the manuscript (line 222).

Line 218-221. This is unconventional in my opinion. The mere use of meta-analysis is that it gives you an overall estimate from estimates that are of known sampling variance (determined by N). Did you assess publication bias?

Our response: This complementary analysis was in fact requested by a previous referee who was concerned about a lack of power due to the high number of studies with a low sample size. We have kept this analysis in the revised version of the manuscript but we are willing to remove it if the editor thinks it is necessary. We tested for but did not find evidence of any publication bias (see Section 6 of the material and method and lines 311-316 in the result section of the revised manuscript).

L226. For many species it will be impossible to distinguish juveniles from adults I would assume?

Our response: To distinguish between juveniles and adults, we performed an intensive literature search to find the age at first reproduction. We were able to retrieve this information for all species included in our dataset (see DataSpecies_ESM and line 191-192 in the revised manuscript).

L237-240. I think you mentioning dynamics quite often in the introduction put me on the wrong foot that you were actually looking at differential shortening rate between the sexes, but you are not. I would make this more explicit.

Our response: We apologize for the lack of clarity in our initial manuscript. As mentioned in a previous comment, we have now specified in the introduction that we were interested in analysing sex differences in mean telomere length to avoid any confusion (lines 121 and 123).

Outlier. I am not convinced this adds a lot really, maybe to try and prove the null hypothesis (which is a fallacy as well), also a bit strange you rely on metafor her to provide you the residual and not MCMCglmm.

Our response: The outlier diagnostic is a common procedure in meta-analysis [16,17]. In addition to this diagnostic, we performed our analyses with and without the effect size considered as an outlier and we reported and discussed both results. We used metafor because the outlier diagnostic is implemented in this package but not in MCMCglmm.

L310. In this model correcting for the one most explanatory variable in your model (gel versus qPCR) what was the estimate of the sex effect at the intercept? Positive, if also negative you might have a sex effect within gel only. Not that it matters much but worth reporting.

Our response: In our model the intercept was set to 0 to estimate a direct effect size for each method. When we tested the TRFI against the qPCR methods, we did not find any statistically significant difference between methods (TRFI vs qPCR estimate \pm 95% CI = 0.31 \pm 0.33).

L315, rephrase 'will be'.

Our response: This has now been corrected (line 329).

L315. This effect cannot really be appreciated without knowing what the intercept is in this model. Needs presenting and interpreting.

Our response: The intercept of the model is: 0.01967, HPDI = [-0.42979:0.47734]. The model here was $g \sim \log(\text{Male body mass}) + \log(\text{Female body mass})$ following Freckleton et al. [18] to avoid performing stats on stats (i.e. using residuals).

L317. I am not very convinced by this outlier analysis. Point of meta-analysis is to quantitatively summarise all data available. Yes worth mentioning this hinges on the extremes and driven by one datapoint (partially). I do not think it is proof this relationship is driven by the outlier or that this point is an actual outlier. What is it an outlier in? In biology? In the sampling variance/error distribution? You won't be able to distinguish those options.

Our response: From a statistical point of view, this point can be confidently interpreted as an outlier (i.e. with a studentized deleted residuals of 4.05 vs. <1.96 expected for 95% of the data points), which could influence the outcome of the analyses (high value of DFFITS and Cook's distance). As mentioned above, the outlier diagnostic is a common procedure in meta-analysis [16,17] and it is thus important to perform our analyses with and without this effect size as well as to present and discuss results obtained with and without it. It appears that this effect size drives the relationship between sex differences in TL and SSD. For the sake of transparency, it is thus very important to mention this aspect. That being said, we agree with the reviewer that from a biological viewpoint, we cannot assess whether this data point is an outlier or not. We thus added a discussion about results including that data point and made explicit that we cannot currently distinguish what is the real outcome (see lines 370-376 in the revised manuscript)

L319. Shouldn't sexual dimorphism in size be logged?

Our response: As mentioned on a previous comment, Sexual Size Dimorphism were assessed using log-scaled male and female body mass as covariates in the model. Mass was log-transformed because a linear relationship between male and female mass should only occur on a log-scale based on allometric rules (which is also the case for the age at first reproduction). This information is reported in Supplementary Methods and Results and is now specified in the main text of the revised version of the manuscript (line 192).

L331. How does the data compare in terms of effect size? Do you detect a similar effect size, but your precision is just lower using a lower overall sample size than in human studies? Are these human studies age-corrected?

Our response: We used a different statistical approach to compute effect size compared to that used by Gardner et al. in their work (Hedge's g in our meta-analysis against a standardized regression coefficient in the analysis of Gardner et al [5]), which prevents us

to compare reliably the estimations. Moreover, effect sizes estimated by Gardner et al. were age-corrected.

L338-340. I do not see this result or test of this in your result section?

Our response: This is just an illustration of our results. No statistical test is associated.

L351-353. The relationship did not disappear, if you test the resulting slope against each other with and without the outlier that wont be different I don't think. I do not think your interpretation of outlier analysis is correct.

Our response: We agree with the referee's comment, that the word 'disappear' was not the most relevant. We have reworded the sentence that now reads 'However, this relationship was non-statistically significant...' (line 374).

L369-371. You will need to review the ages of the human studies that claim sex differences for this. How does this fit with the relationship between TL and mortality being lower in older humans (or completely gone in old people..).

Our response: In the meta-analysis we quoted (*i.e.* [5]), effect sizes were age-corrected. However, as mentioned in our work most studies in humans have been performed on adults and on elderly people, which contrast markedly with studies on animals in the wild more focused on young and prime-aged individuals, which might explain why we did not detect any sex differences in our meta-analysis.

L393-398. This is wrong. Sample size included should not influence I^2 .

Our response: Respectfully, we disagree with the referee on that point.

Following [19], we calculated I^2 as $I^2 = \frac{\sigma_u^2}{\sigma_u^2 + \sigma_m^2}$ with σ_u^2 the between-study variance and σ_m^2 the within-study variance (also called the sampling error variance). The within-study variance σ_m^2 is estimated from the study-specific sampling error variance Vg with $Vg = \frac{nM+nF}{nM*nF} + \frac{g^2}{2(nM+nF)}$ where nM and nF are the sample size for males and females, respectively, and g the value of our effect size, the Hedge's g , following [11]. Therefore, it results that the I^2 is clearly influenced by the study-specific sample sizes.

L402-403. You want to compare effect size. I am unclear as to why you are not formally comparing that to the human effect. It might actually not differ from that at all. You have a testable hypothesis here and a meta-analysis to quantitatively compare effects and their confidence/credibility intervals.

Our response: This is a really interesting point. However, as mentioned above, due to differences in methodology, it was not possible to compare directly the results from our meta-analysis and the results from the meta-analysis performed across human populations [5].

L405-407. Again sample size included should not influence I^2 . I think you misunderstand this part of the meta-analytic methodology?

Our response: Please see our response above.

I think your discussion needs an appreciation of telomere attrition over absolute telomere length differences and what they tell us about the biology of ageing. For example recent comparative analyses show telomere shortening related to a species' lifespan. Yet telomere shortening is less predictive of mortality in humans than absolute telomere length is. For other vertebrates I would not know what the current consensus is.

Our response: In the third meta-analysis we performed, we used age-corrected adult telomere length to calculate the sex difference in mean telomere length (See section 5 of the Material and Method). However, results from this analysis were qualitatively the same as those of our main analysis (See Table S4 and Table S5), which leads to the conclusion that there is no sex difference in telomere length, even when the age-specific changes in telomere length are taken into account. This is now mentioned in the discussion (lines 344-346). Regarding the relationship between telomere shortening or absolute telomere length and ageing in non-human vertebrates, there is no consensus yet in the literature. Indeed, both mean telomere length [3,20,21] and telomere attrition rate [21–23] have been related to mortality and lifespan.

References:

1. Boonekamp JJ, Simons MJP, Hemerik L, Verhulst S. 2013 Telomere length behaves as biomarker of somatic redundancy rather than biological age. *Aging Cell* **12**, 330–332. (doi:10.1111/ace1.12050)
2. Rudolph KL, Chang S, Lee H-W, Blasco M, Gottlieb GJ, Greider C, DePinho RA. 1999 Longevity, Stress Response, and Cancer in Aging Telomerase-Deficient Mice. *Cell* **96**, 701–712. (doi:10.1016/S0092-8674(00)80580-2)
3. Wilbourn RV, Moatt JP, Froy H, Walling CA, Nussey DH, Boonekamp JJ. 2018 The relationship between telomere length and mortality risk in non-model vertebrate systems: a meta-analysis. *Philosophical Transactions of the Royal Society B: Biological Sciences* **373**, 20160447. (doi:10.1098/rstb.2016.0447)
4. Armanios M, Alder JK, Parry EM, Karim B, Strong MA, Greider CW. 2009 Short telomeres are sufficient to cause the degenerative defects associated with aging. *Am. J. Hum. Genet.* **85**, 823–832. (doi:10.1016/j.ajhg.2009.10.028)
5. Gardner M *et al.* 2014 Gender and telomere length: systematic review and meta-analysis. *Exp. Gerontol.* **51**, 15–27. (doi:10.1016/j.exger.2013.12.004)
6. Factor-Litvak P, Susser E, Kezios K, McKeague I, Kark JD, Hoffman M, Kimura M, Wapner R, Aviv A. 2016 Leukocyte Telomere Length in Newborns: Implications for the Role of Telomeres in Human Disease. *Pediatrics* **137**, e20153927. (doi:10.1542/peds.2015-3927)
7. Perner S *et al.* 2003 Quantifying Telomere Lengths of Human Individual Chromosome Arms by Centromere-Calibrated Fluorescence in Situ Hybridization and Digital Imaging. *Am J Pathol* **163**, 1751–1756.
8. Barrett ELB, Richardson DS. 2011 Sex differences in telomeres and lifespan. *Aging Cell* **10**, 913–921. (doi:10.1111/j.1474-9726.2011.00741.x)
9. Xirocostas ZA, Everingham SE, Moles AT. 2020 The sex with the reduced sex chromosome dies earlier: a comparison across the tree of life. *Biology Letters* **16**, 20190867. (doi:10.1098/rsbl.2019.0867)
10. Hemann MT, Greider CW. 2000 Wild-derived inbred mouse strains have short telomeres. *Nucleic Acids Res.* **28**, 4474–4478. (doi:10.1093/nar/28.22.4474)
11. Koricheva J, Gurevitch J, Mengersen K. 2013 *Handbook of Meta-analysis in Ecology and Evolution*. Princeton University Press.
12. Dormann CF *et al.* 2013 Collinearity: a review of methods to deal with it and a simulation study evaluating their performance. *Ecography* **36**, 27–46. (doi:10.1111/j.1600-0587.2012.07348.x)
13. Wilbourn RV *et al.* 2017 Age-dependent associations between telomere length and environmental conditions in roe deer. *Biology Letters* **13**. (doi:10.1098/rsbl.2017.0434)

14. Gardner JP *et al.* 2007 Telomere dynamics in Macaques and humans. *Journals of Gerontology Series A-Biological Sciences* **62**, 367–374. (doi:10.1093/gerona/62.4.367)
15. Brooks SP, Gelman A. 1998 General Methods for Monitoring Convergence of Iterative Simulations. *Journal of Computational and Graphical Statistics* **7**, 434–455. (doi:10.1080/10618600.1998.10474787)
16. Hedges LV, Olkin I. 2014 *Statistical Methods for Meta-Analysis*. Academic Press.
17. Rosenthal R. 1995 Writing Meta-analytic reviews. *Psychological Bulletin* **118**, 183–192. (doi:10.1037/0033-2909.118.2.183)
18. Freckleton RP, Harvey PH, Pagel M, Losos AEJB. 2002 Phylogenetic Analysis and Comparative Data: A Test and Review of Evidence. *The American Naturalist* **160**, 712–726. (doi:10.1086/343873)
19. Nakagawa S, Santos ESA. 2012 Methodological issues and advances in biological meta-analysis. *Evolutionary Ecology* **26**, 1253–1274. (doi:10.1007/s10682-012-9555-5)
20. Bichet C, Bouwhuis S, Bauch C, Verhulst S, Becker PH, Vedder O. 2020 Telomere length is repeatable, shortens with age and reproductive success, and predicts remaining lifespan in a long-lived seabird. *Molecular Ecology* **29**, 429–441. (doi:10.1111/mec.15331)
21. Bize P, Criscuolo F, Metcalfe NB, Nasir L, Monaghan P. 2009 Telomere dynamics rather than age predict life expectancy in the wild. *Proceedings of the Royal Society B: Biological Sciences* **276**, 1679–1683. (doi:10.1098/rspb.2008.1817)
22. Tricola Gianna M. *et al.* 2018 The rate of telomere loss is related to maximum lifespan in birds. *Philosophical Transactions of the Royal Society B: Biological Sciences* **373**, 20160445. (doi:10.1098/rstb.2016.0445)
23. Whittimore K, Vera E, Martínez-Nevado E, Sanpera C, Blasco MA. 2019 Telomere shortening rate predicts species life span. *PNAS* **116**, 15122–15127. (doi:10.1073/pnas.1902452116)